# The Multifaceted Importance of Amphibians: Ecological, Biomedical, and Socio-Economic Perspectives

**DOI:** 10.3390/biology15010098

**Published:** 2026-01-02

**Authors:** Buddhika Wickramasingha, Josh West, Bellanthudawage Kushan Aravinda Bellanthudawa, Michael P. Graziano, Thilina D. Surasinghe

**Affiliations:** 1Centre for Sustainability Solutions, University of Kelaniya, Kelaniya 11600, Sri Lanka; buddhikasa10@gmail.com; 2Department of Biological Sciences, Bridgewater State University, Bridgewater, MA 02325, USA; j2west@student.bridgew.edu; 3Department of Civil and Environmental Engineering, Faculty of Engineering, Monash University, 21 College Walk, Clayton, VIC 3168, Australia; kushan.bellanthudawage@monash.edu; 4Biology Department, University of Hartford, West Hartford, CT 06117, USA; migrazian@hartford.edu

**Keywords:** amphibian conservation, biodiversity loss, bioindicators, ecosystem services, environmental change

## Abstract

Amphibians—including frogs, toads, salamanders, and caecilians—are declining at an alarming pace globally, raising serious concerns for biodiversity and ecosystem health. Scientists estimate that over 40% of amphibian species are now threatened with extinction. This rapid loss necessitates an integrated, multi-faceted conservation mechanism for amphibians. Our review synthesizes the importance of amphibians from biodiversity, ecosystem health, and human well-being perspectives. Amphibians play important roles in regulating insect pests, nutrient cycling, and serve as prey for numerous species. They also have medical value because their skin contains compounds that may lead to new medicines. Unfortunately, habitat destruction, climate change, pollution, invasive species, and diseases are driving many populations to decline. We gathered current knowledge about the roles amphibians play in the wild and explained how their decline can impact ecosystems. We also highlighted how new technologies such as satellite remote sensing and geo-spatial sciences as well as citizen science can help track these changes and support conservation efforts. Importantly, we call for improved public awareness and educational programs, use of indigenous knowledge, and stronger policies that recognize amphibians not just as endangered species, but as vital parts of functioning ecosystems. Protecting these sensitive animals is not only about preserving amphibians but also about protecting nature’s balance and our own health.

## 1. Introduction

Amphibians are among the most imperiled vertebrate taxa, with more than 40% of species threatened globally, a crisis that demands urgent, evidence-based conservation and management responses [1]. However, current conservation measures often fail to leverage the ecological and socioeconomic value of amphibians, weakening the case for sustained conservation investment. Amphibians are integral to ecosystem functioning, including regulating disease vectors, facilitating nutrient cycling, and supporting energy transfer across aquatic and terrestrial systems [2,3]. They also provide economically vital services through biocontrol and bioprospecting [2]. This disconnect between their ecological importance and conservation attention stems in part from a fragmented knowledge base [4,5]. While many studies have addressed specific ecological functions or economic values [2,3], no recent synthesis integrates these dimensions to guide comprehensive, cross-sectoral conservation planning. Here, we bridge this gap by systematically reviewing the multidimensional roles of amphibians as (1) ecological engineers; (2) biomedical innovators; (3) cultural-socioeconomic assets, and (4) indicators of environmental health. By integrating these domains, we strengthen the case for amphibian-focused conservation policy and reframe their protection as an investment in both ecosystem integrity and human well-being. Our review provides comprehensive evidence to elevate amphibians within global conservation agendas including the Convention on Biological Diversity, Nature-Based Solutions frameworks, and the United Nations Sustainable Development Goals by demonstrating their irreplaceable contributions to the environment and society.

### 1.1. Global Patterns of Amphibian Diversity

Amphibians are an ancient, remarkably diverse, monophyletic vertebrate class, currently comprising 8981 described species across three taxonomic orders: Anura (frogs and toads), Caudata (salamanders), and Gymnophiona (caecilians) [6]. Emerging over 350 million years ago during the Devonian period (419–359 million years ago), amphibians mark a pivotal transition in tetrapod evolution from an aquatic to terrestrial mode of life [7,8]. Since then, they have undergone extensive adaptive radiation, colonizing every continent except Antarctica with a niche diversification that spans terrestrial, aquatic, scansorial, and fossorial realms throughout both the Old and New Worlds [9,10,11]. This ecological breadth extends along elevational gradients from lowlands to montane regions in tropical, subtropical, and temperate biomes, facilitated by remarkable physiological, behavioral, and morphological adaptations [10,12]. Global amphibian species richness exhibits pronounced heterogeneity, with tropical and subtropical regions supporting hyperdiverse and highly endemic assemblages adapted to warm, wet climates and complex topography [8,10,11]. The convergence of high species richness, narrow ecological specialization, and intrinsic vulnerability highlights the urgent need to elucidate the diverse functional roles amphibians play both locally and globally. Beyond their ecological distinctiveness and evolutionary significance, amphibians occupy a multifaceted functional niche. Consequently, understanding the ecological and functional importance of amphibians is not merely an academic pursuit; it is fundamental to assessing the cascading consequences of their ongoing global declines and informing effective conservation strategies aimed at preserving ecosystem integrity and services [13,14].

### 1.2. Global Hotspots of Amphibian Diversity

Recent biogeographic analyses underscore the importance of historical geomorphic processes, climatic refugia, and habitat heterogeneity in shaping present-day amphibian diversity. In the western Amazon, for instance, geological uplift and rainfall gradients have created discrete microhabitats fostering high endemism and range-restricted species [15,16]. Isolated cloud forests in Honduras, Guatemala, and Costa Rica harbor endemic genera such as *Craugastor* and *Bolitoglossa* [17,18]. The Neotropical region contains the highest amphibian richness globally, where Brazil, Colombia, Ecuador, and Peru collectively harbor more than 3000 species, including a significant proportion of endemics [13,14]. Brazil alone has documented over 1000 species, with many recent descriptions resulting from field surveys along with taxonomic revisions, molecular phylogenetics, and bioacoustic analyses [19]. Within this region, the Andean foothills and Amazon Basin represent an amphibian diversity epicenter where elevational gradients and climatic stability have promoted allopatric speciation and ecological diversification [13,14,19]. In North America, the Appalachian Mountains are notable for their exceptional diversity of Plethodontid (lungless) salamanders [20,21].

Africa’s amphibian diversity is centered in the Guineo–Congolian rainforests, the Albertine Rift, and Madagascar. The island of Madagascar supports more than 300 species, with endemism exceeding 99% resulting from the islands’ long-term geographic isolation and microclimatic heterogeneity [14,22]. In East Africa, Ethiopian Highlands and the Eastern Arc Mountains are increasingly recognized for their richness and endemism [23,24].

In Asia, the Indomalayan realm including the Western Ghats, Sri Lanka, Southeast Asia, and southern China has emerged as a major hotspot for amphibian discovery [25,26]. Similarly, Annamite Mountains and Sundaland region of Southeast Asia are characterized by high beta-diversity and cryptic species complexes [27,28]. The Australasian region, especially eastern Australia and New Guinea, harbors distinctive radiations of microhylids, ground-dwelling myobatrachids, and the iconic, now-extinct gastric-brooding frogs (*Rheobatrachus* spp.) [29].

### 1.3. Urgency of Conservation: Global Amphibian Decline and the Loss of Ecosystem Services

Since the 1980s, global scientific consensus has reported that amphibians are declining at an alarming rate [30]. The second Global Amphibian Assessment (2023) demonstrates that 41% of amphibian species are threatened with extinction surpassing birds (13%) and mammals (27%) while 49% exhibit documented population declines [1,31]. This crisis, tracked through decades of IUCN Red List evaluations, emphasizes an urgent imperative for evidence-based conservation [32,33] and necessitates immediate integration of amphibian-centric policies into global biodiversity frameworks. In contrast to many vertebrate declines that are closely associated with proximate anthropogenic pressures, amphibian declines frequently persist even within protected areas, underscoring the vulnerability of conservation lands to broadcast threats; similar patterns, though less well documented, have been observed in certain bird and reptile populations [34,35]. Multiple, interacting drivers are responsible for these declines. Habitat destruction and fragmentation, urban development, emerging infectious diseases, climate change, chemical pollution, and biological invasions remain the most widespread pressures affecting amphibians worldwide [36,37,38,39].

The ecological consequences of amphibian declines extend far beyond species loss. Amphibians act as invaluable ecological agents, with larvae influencing productivity and decomposition in freshwater systems, adults regulating terrestrial invertebrate populations, and their dual role as both predator and prey supporting trophic stability and nutrient dynamics [40,41,42]. Amphibian population declines and extirpation can trigger cascading ecological outcomes. Long-term studies in Central America, for example, have linked these losses to marked shifts in stream ecosystem structure and function, including increased algal biomass, altered detrital processing, and reduced predator densities [40,43]. Amphibians provide critical biocontrol services, and the collapse of amphibian communities in Central America has resulted in increased mosquito populations and a subsequent rise in human malaria incidence [44,45]. In agricultural landscapes, loss of native frog species has been linked to increased pest outbreaks and declining crop yields [46,47]. Moreover, declines in amphibians can disrupt food chains that support higher trophic levels. For instance, widespread amphibian extinctions in Panama reduced snake species richness by 85% [48,49].

Beyond ecological functions, the continuous reduction in amphibian diversity shows a loss of ecosystem services that support human health, economic livelihoods, and cultural practices. Their skin-derived compounds hold potential for developing new treatments for infectious diseases, pain management, and even regenerative therapies [50,51]. Each extinction represents not only an irreversible loss of biodiversity, but also the potential loss of as-yet-undiscovered medical treatments. The urgency of amphibian conservation, therefore, cannot be overstated [52]. Given their integral roles in ecosystem processes, amphibian conservation must encompass the maintenance of ecosystem functioning, the stability of ecological services, and environmental resilience.

### 1.4. Methodological Framework of This Review

This review provides a comprehensive understanding of the multifaceted importance of amphibians by synthesizing current research across ecological, biomedical, and socio-economic domains. We synthesize peer-reviewed literature, international reports, and conservation datasets published primarily between 2000 and 2024, with in-depth analyses on studies from the last five years. Sources were identified using structured keyword searches in academic databases such as Web of Science, PubMed, and Scopus. Key search terms included “amphibia” in combination with “ecosystem services,” “ecosystem functions,” “ecological processes,” “ecological role,” “ecological importance,” and “bioindicators.” The selected literature was analyzed qualitatively to identify recurring themes, novel findings, conservation implications, and knowledge gaps. By contextualizing these dimensions within the global amphibian decline, we highlight the urgent need for interdisciplinary conservation strategies and offer a framework for integrating amphibians into broader sustainability, health, and biodiversity agendas.

## 2. Amphibians as Key Players in Ecosystem Stability

### 2.1. Amphibians as Foundational Species

Amphibians function as foundational species, driven in part by their high aggregate biomass, which influences their roles in the bioturbation of aquatic sediments, nutrient transfer between habitats, and the creation or modification of microhabitats (Figure 1) [53,54,55]. Amphibians link aquatic and terrestrial energy pathways through their biphasic life cycles [56]. These cross-system linkages position amphibians as energy vectors, facilitating material flow between aquatic and terrestrial food webs [41]. Larvae consume periphyton, detritus, and microbial biofilms in freshwater systems, thereby altering algal productivity and nutrient stoichiometry, while adults prey on invertebrates while serving as a prey base for other consumers, mediating trophic cascades in terrestrial habitats [57].

In many ecosystems, amphibians predominate among other vertebrates in biomass; this can support broader community stability [41]. In temperate forests, terrestrial salamanders (genus *Plethodon*) predominate in vertebrate biomass (~1650 g of wet weight/ha) [58,59], often exceeding that of birds and mammals (Table 1). This numerical abundance amplifies their functional roles such as processing invertebrate prey thereby contributing to ecosystem energy budgets, modifying soil microhabitats through burrowing, and retaining moisture [60]. As secondary consumers, terrestrial salamanders regulate detritivore communities, thereby contributing to decomposition, carbon cycling, and nutrient retention [41].

Tadpoles play a foundational role in many freshwater ecosystems, particularly in the tropical regions of the Amazon and Southeast Asia [61]. Their exceptionally high densities in some ecosystems allow them to exert strong influences on benthic structure, primary production, and nutrient fluxes [2,62]. Further, through intensive grazing and sediment-feeding, tadpoles regulate algal and periphyton communities and shift assemblages toward grazing-resistant taxa [63,64]. Simultaneously, they modulate sediment dynamics via bioturbation and particle ingestion, while decreasing total benthic sediment loads, including organic and inorganic fractions, by 41–43% [65]. These processes increase nutrient availability and facilitate higher densities of other consumers in the ecosystems [62]. Such sediment-feeding activities can also create competition with sediment-feeding fish while facilitating others that rely on exposed aquatic substrates [66]. Additionally, mass metamorphic emergence in both tropical and temperate systems transfers substantial biomass (up to 2500 kg/ha/year) from aquatic to terrestrial ecosystems [67].

**Table 1 biology-15-00098-t001:** Comparison of biomass contribution of amphibians vs. other vertebrates in different ecosystems.

Ecosystem Type	Region/ Country	Amphibian Biomass (kg/ha)	Bird Biomass (kg/ha)	Mammal Biomass (kg/ha)	Measurement Method	References
Temperate Forest	New Hampshire, USA	3.7–4.6	~1.3	~0.3	Pitfall traps, dry mass	[68]
Subtropical Wet Forest	Puerto Rico	0.41–2.68 (wet weight)	NR	NR	Visual encounter, wet mass	[69,70]
Cloud Forest	Ecuador	1.2	0.08	0.05	Visual encounter surveys	[70,71]
Temperate Deciduous	Ontario, Canada	1.0–2.8	NR	NR	Quadrat dry mass sampling	[2,72]
Montane Stream Forest	Costa Rica	5.6	NR	NR	Visual counts, wet biomass	[73]
Tropical Forest	Puerto Rico	~4.0 (C/N)	NR	NR	Nutrient cycling residue mass	[74,75]
Rainforest Streams	Panama	0.15–3.6 μg NH_4_^+^/h	NR	NR	Nutrient excretion (lab)	[43,75]
Mediterranean Springs	Spain	High density reported	NR	NR	Breeding & larval density counts	[76]
Mountain Wetlands	Colombia	20,000 individuals/ha	NR	NR	Density counts	[43]
Temperate Forest Streams	USA (Appalachians)	~0.8–1.5	~1.0	~0.2	Mark-recapture, dry mass	[77]
Lowland Rainforest	French Guiana	2.4	0.5	0.6	Pitfall and quadrat counts	[78]
Agricultural Wetland	India	High relative density	NR	NR	Quadrat surveys	[79]
Floodplain Forest	Peru	Not quantified	NR	NR	Vocalization and sighting counts	[67,80]
Temperate Swamp	Canada	~1.5	~0.6	~0.4	Pitfall and call index	[80]
Isolated Wetlands	Maine, USA	3.2–6.4 (export)	NR	NR	Drift fences, pitfall traps	[81]

Note: NR = Not Reported.

### 2.2. Nutrient Cycling and Energy Flow Mediated by Amphibians

Feeding behavior (e.g., grazing) of larval anurans accelerates organic matter decomposition and nutrient mineralization (e.g., nitrate, orthophosphate), modifying aquatic nutrient composition and concentration [82,83]. They also immobilize phosphorus and calcium during development, with additional sequestration into larval skeletons, further contributing to shifts in nutrient composition [68]. Amphibians with biphasic life cycles are critical trophic links mediating fluxes of nitrogen, phosphorus, and carbon between aquatic and terrestrial ecosystems [41,57]. While post-metamorphic amphibian dispersal from wetlands into upland habitats illustrates cross-ecosystem connectivity, the reciprocal flow of nutrients into aquatic systems via egg deposition also constitutes a significant, yet often underrecognized, ecological subsidy. For example, glass frogs at El Copé, Panama, deposit 0.64 egg masses/m^2^ of stream habitats each breeding season, representing a notable input of organic matter into aquatic systems [40,84]. Larval excretions by glass frogs account for ~7% of bulk ammonium uptake in Neotropical streams, highlighting their role in nitrogen cycling and nutrient dynamics [81,85,86].

Terrestrial plethodontid salamanders (*Plethodon* spp.) act as concentrated reservoirs of critical limiting nutrients (retaining 7–15% of forest elemental fluxes, e.g., 0.6 kg Nitrogen/ha, 0.15 kg Calcium/ha), functioning as standing stock for calcium, sulfur, phosphorus, magnesium, and nitrogen within the system [83,87]. Their chitin-free carcasses counteract soil acidification-driven calcium/magnesium depletion upon decomposition [88]. Amphibians exhibit high assimilation efficiency and low metabolic expenditure (~60% of ingested energy into new tissue) due to their ectothermic physiology—characterized by reliance on external heat sources—which minimizes endogenous metabolic costs, resulting in high annual tissue production and standing biomass (e.g., *Plethodon serratus*: 5769–10,195 g·ha^−1^ wet mass, 5287–9343 g·ha^−1^ protein) [41]. This high-energy, nutrient-dense biomass represents a crucial resource for higher trophic levels.

In terrestrial ecosystems, adult amphibians function as both consumers and nutrient transporters. Predation on invertebrates, many of which are herbivorous pests, contributes indirectly to stabilizing aboveground vegetation biomass while enhancing nutrient transfer from prey into the soil via excretion [89]. Their feces are often hotspots of microbial activity contributing to nitrogen assimilation and organic matter turnover [90]. In temperate zones, terrestrial salamanders significantly alter nitrogen cycling in forest soils through detritivore consumption and excretion of nitrogenous waste [68,89]. In tropical systems, amphibian larvae mediate nutrient dynamics via herbivory, cannibalism, and sediment disturbance [86,91]. Through the deposition of nutrient-rich urine, feces, and carcasses, the arboreal frog (*Eleutherodactylus coqui)* elevates the concentration of biologically critical nutrients in subtropical forest-floor ecosystems. This vertical subsidy transfers dissolved and particulate nutrients from the canopy to the detrital layer, and the rapid return of nutrients in highly labile forms accelerates nutrient cycling and stimulates microbial and plant productivity [74].

### 2.3. Trophic Contributions and Food Web Integration

Amphibians’ dual role as consumers of invertebrates and as a key prey source for higher trophic levels generates trophic complexity and enhances energy transfer efficiency across ecosystems [41,89]. In their larval stages, amphibians exploit a range of trophic niches depending on habitat and species. For instance, many tadpoles are primary consumers, feeding on algae, detritus, and periphyton, thereby mediating bottom–up energy pathways and controlling basal resources in aquatic systems [86,90]. Others, particularly in nutrient-poor systems, exhibit carnivory and cannibalism, triggering trophic dynamics in resource-limited environments [86,92]. Grazing tadpoles exert strong top-down control in tropical headwater streams by reducing algal biomass, modifying algal community composition, and minimizing sediment accumulation through ingestion and bioturbation [62].

Adult amphibians are generally insectivorous. In agroecosystems, their predation significantly reduces herbivorous pest and zoonotic disease vector (e.g., mosquitoes) populations [41,89], providing pest control services [46]. In temperate forest-floor ecosystems, terrestrial salamanders exert top–down control on soil mesofauna and significantly reduce their abundance, including oribatid and non-oribatid mites and Onychiurid Collembola [93]. Amphibians can restructure detritus-based food webs by reducing macroinvertebrate detritivores and enhancing mesofaunal populations through predation and nutrient subsidies, which may alter invertebrate community composition [94]. Through predation, terrestrial salamanders have suppressed the abundance of dominant key large-bodied leaf-fragmenting invertebrates (millipedes, fly and beetle larvae, mollusks, spiders, and flying insects), slowing leaf-litter decomposition in temperate forested ecosystems [95].

Amphibians, especially their eggs and larvae, constitute a substantial prey base for both aquatic and terrestrial predators, enhancing trophic connectivity by linking aquatic and terrestrial systems [2,42]. In certain food webs, amphibians sustain predator populations during seasonal shifts when prey is scarce, thereby contributing to the stability of trophic dynamics [96]. High amphibian densities and biomass in isolated wetlands (e.g., 38,612 individuals/ha and 159 kg/ha/year at Carolina Bay wetlands) highlight their capacity to export substantial energy to terrestrial ecosystems post-metamorphosis [97].

Amphibians exhibit highly efficient conversion of ingested food into body mass, with minimal losses to metabolic heat, resulting in a greater proportion of energy being stored as biomass, substantially increasing their caloric density [98]. Consequently, amphibians serve as energetically valuable prey, functioning as bioaccumulators and long-lived energy reservoirs, particularly in resource-limited ecosystems [41,68]. Seasonal movements of breeding adults and metamorphosed individuals facilitate energy transfer between aquatic and terrestrial systems, providing high-quality trophic subsidies for terrestrial predators [82].

### 2.4. Ecosystem Engineering

Fossorial amphibians function as ecosystem engineers by physically altering soil environments through burrowing and estivation. Species such as *Rhinophrynus dorsalis* (Mexico and Central America), *Nasikabatrachus sahyadrensis* (India), *Scaphiopus* spp (North America), and various caecilians contribute to soil bioturbation, enhancing aeration, water infiltration, nutrient redistribution, and microbial colonization [40,42,89,99]. These modifications increase soil porosity, restore soil functions, loosen compacted soil layers, and facilitate root development and microbial activities. In arid and semi-arid biomes, these subterranean activities improve moisture retention in desiccated substrates, thereby enhancing habitat quality for other organisms [100].

Additionally, amphibians influence habitat microtopography and localized hydrology through nest construction. Foam-nesting frogs such as *Leptodactylus* spp. excavate shallow depressions that retain moisture and function as nutrient sinks [41,99]. In leaf-litter-dominated and montane forest systems, terrestrial salamanders like *Plethodon cinereus* stir surface litter and expose underlying soil while foraging, promoting microbial respiration and decomposition [41,68]. These modifications enhance nutrient cycling, oxygen availability, and soil heterogeneity, benefiting fungi, detritivores, and plant roots alike. Though often overlooked compared to macrofauna, amphibians contribute subtly yet critically to belowground biodiversity, making them key agents in soil ecosystem function and resilience [89].

## 3. Biomedical Significance of Amphibians

### 3.1. Antimicrobial Peptides and Their Medical Applications

Amphibians possess an exceptional array of antimicrobial peptides (AMPs) in their skin secretions, which serve as a vital component of their innate immunity [101,102]. These peptides, which are generally amphipathic and cationic, interact with microbial membranes to exert potent antimicrobial, antiviral, antifungal, anticancer, antioxidant, and wound-healing properties [103]. Over the past few decades, research has identified and characterized dozens of such peptides, each with unique sequences, activities, and therapeutic potential [104]. The wide taxonomic diversity of amphibians has enabled the discovery of numerous AMPs, including magainins, dermaseptins, brevinins, esculentins, bombesins, temporins, phylloseptins, and novel families such as cruzioseptins and syphaxins, spanning across the genera *Xenopus*, *Phyllomedusa*, *Odorrana*, *Leptodactylus*, *Hylarana*, and *Bombina* [105,106,107,108,109,110,111,112,113,114,115,116,117,118].

One of the earliest characterized AMPs, magainin, was isolated from *Xenopus laevis* and has since become a prototype for synthetic analogs such as pexiganan. These peptides form toroidal pores in microbial membranes, leading to cytolysis of Gram-negative and Gram-positive bacteria, as well as fungi such as *Candida albicans* [119,120]. Pexiganan reached late-phase clinical trials for diabetic foot ulcer treatment, demonstrating the translational viability of amphibian-derived peptides [121]. Similarly, dermaseptins from *Phyllomedusa bicolor* show significant efficacy against both microbial pathogens and tumor cells [122,123]. Besides classical AMPs, recent discoveries from *Odorrana*, *Rana*, and *Hylarana* genera have unveiled multifunctional peptides with antioxidant and regenerative effects. Peptides such as OA-GL21 and Cathelicid-OA1 promote wound healing by stimulating TGF-β1 expression and enhancing glutathione (GSH) levels, thereby accelerating tissue repair [101,124]. Andersonin-G1 and APBSP, identified in Rana species, have been shown to scavenge free radicals such as DPPH and superoxide, thereby reducing oxidative stress and highlighting an emerging mechanism in antimicrobial defense and chronic disease prevention [101,125]. Importantly, amphibian peptides also exhibit insulinotropic and anticancer properties. Brevinins from *Lithobates pipiens*, *Hylarana guntheri*, and *Odorrana* species stimulate insulin release in pancreatic β-cells, offering therapeutic leads for type 2 diabetes mellitus [126]. Peptides like Hymenochirin-1B and XLAsp-P1 induce apoptosis in tumor cells by disrupting mitochondrial membranes, without affecting healthy cells, indicating selective cytotoxicity suitable for oncology applications [127,128].

A subset of peptides, bombesins, found in *Bombina* and *Boana* species, influence hormonal pathways by stimulating gastrin release. Moreover, peptides like magainin II demonstrate spermicidal effects and are being investigated as dual-action antimicrobial contraceptives [127,129]. These multifunctional properties reinforce the biomedical significance of amphibian AMPs across diverse medical domains.

More than 50 well-documented peptides, alkaloids, and other bioactive compounds have been extracted from amphibian skin (Table 2). Research studies are now shifting towards optimizing AMP pharmacokinetics, enhancing peptide stability in physiological fluids, and developing nanoparticle-based or liposomal delivery systems [101,130]. As the global threat of antimicrobial resistance intensifies, amphibian AMPs originating from millions of years of evolutionary refinement offer an invaluable, underexplored biochemical arsenal. Their integration into pharmaceutical pipelines represents not only a biomedical opportunity but also a strong argument for the conservation of amphibian biodiversity as a reservoir of natural therapeutics.

### 3.2. Regenerative Medicine

Among amphibians, salamanders stand as the most remarkable models of regeneration. They possess the ability to completely regenerate complex structures such as limbs, spinal cords, eyes, heart tissue, and even portions of the brain and lungs following injury [162]. These phenomena are not just of evolutionary interest but also represent a significant reservoir of biological insights for regenerative medicine, wound healing, and tissue engineering [163]. The foundation of amphibian regeneration lies in their ability to initiate a well-orchestrated cellular response at the site of injury. Upon tissue damage, epidermal cells rapidly migrate to cover the wound, forming an apical epithelial cap, which releases signaling molecules such as fibroblast growth factors, bone morphogenetic proteins [164,165]. Experimental studies in salamanders demonstrate that following limb amputation, differentiated cells such as muscle and cartilage can lose their specialized features and contribute to a pool of proliferating progenitor cells beneath the wound epidermis, forming the blastema [166,167]. The blastema serves as the source of new tissues during regeneration and, unlike mammalian wound healing, it reconstructs the missing structures with minimal scarring by re-establishing developmental programs that resemble those seen in embryogenesis.

One of the pivotal genetic regulators of this process is the msx1 gene, a transcription factor involved in maintaining cells in an undifferentiated state. Groundbreaking work by Odelberg et al. [167] demonstrated that overexpression of msx1 in differentiated mouse myotubes induced dedifferentiation and proliferation, an event previously thought impossible in mammalian cells [167].

More recently, advances in genomics have provided unprecedented insight into the molecular basis of amphibian regeneration [168]. The sequencing of the axolotl (*Ambystoma mexicanum*) genome, a species renowned for its lifelong regenerative ability, revealed numerous regeneration-specific genes and regulatory elements which are absent or not active in mammals [169]. These include upregulated genes related to extracellular matrix remodeling, immune modulation, epigenetic reprogramming [170], and redox homeostasis during limb regeneration.

Another critical component enabling this regenerative process is the amphibian immune response. Unlike in mammals, where inflammation often leads to fibrosis and scarring, amphibians produce a transient and non-destructive inflammatory response [171]. Anti-inflammatory macrophage subtypes dominate the injury site and secrete cytokines and growth factors that favor tissue regeneration over scar formation [172]. Experimental depletion of macrophages in axolotls, for instance, has been shown to completely halt regeneration, underscoring their indispensable role in tissue reprogramming and morphogenesis [171].

In spinal cord injury models, axolotls exhibit complete axonal regrowth and restoration of function, without glial scarring. Studies show that ependymal glial cells in the central canal proliferate and contribute to the regeneration of neurons and glia, with re-expression of neurodevelopmental genes such as sox2 and pax6 [173]. These insights can catalyze research into developing therapies for human spinal injuries, stroke, and neurodegeneration. Furthermore, the ability of amphibians to regenerate heart tissue particularly cardiomyocytes and valves after injury is of great interest for cardiac regenerative therapies [174]. In axolotls, cardiac regeneration occurs without fibrosis and involves proliferation of pre-existing cardiomyocytes rather than stem cell recruitment [175].

### 3.3. Potential Therapeutic Compounds

Amphibians possess one of the richest and most pharmacologically promising chemical arsenals among vertebrates [176]. Beyond AMPs, amphibians secrete a wide spectrum of bioactive molecules ranging from neurotoxins and hormone analogs to antioxidants and antidiabetic peptides with diverse mechanisms of action and therapeutic potential [130,176]. These compounds, primarily synthesized in skin glands as part of innate defense strategies, are increasingly viewed as a biochemical goldmine. One major category of amphibian-derived therapeutics includes vasoactive peptides, particularly bradykinin-related peptides [177]. For instance, phyllokinin, isolated from *Phyllomedusa hypochondrialis*, mimics bradykinin activity, causing endothelium-dependent vasodilation through B2 receptor activation, making it a potential candidate for antihypertensive therapies [178]. Similarly, ranakinestatin-PPF and phylloseptin-L2 stimulate nitric oxide release and smooth muscle relaxation, providing pharmacological leads for managing cardiovascular disorders [117,179]. Regarding diabetes treatment, several peptides stand out due to their insulinotropic effects. Notably, tigerinin-1R, derived from *Hoplobatrachus rugulosus*, enhances glucose-stimulated insulin secretion without cytotoxicity [180]. Amphibians also produce cytotoxic peptides and alkaloids with potential applications in oncology. Hymenochirin-1B, isolated from *Hymenochirus boettgeri,* has been shown to induce apoptosis in human breast and lung cancer cells by disrupting mitochondrial integrity and activating caspase-dependent pathways [181]. Similarly, XLAsp-P1 from *Xenopus laevis* and dermaseptin-PP from *Phyllomedusa palliata* exhibit selective cytotoxicity toward tumor cells while sparing normal cells, an essential criterion for anticancer drug development [138].

Analgesic and neuroactive peptides also feature prominently among amphibian secretions. For example, phyllocaerulein, a caerulein-like peptide found in *Phyllomedusa sauvagii*, exhibits both analgesic and antihypertensive effects [136]. Moreover, compounds such as ceruletide and delotorphin act on neurokinin and opioid receptors, presenting alternatives to traditional pain medications with potentially reduced addiction profiles [182]. Additionally, amphibians also secrete unique peptides like cathelicid-OA1, ranatuerin-2P, and esculentin-2P, which exhibit dual activity in wound healing and microbial inhibition, positioning them as multifunctional therapeutic leads. For instance, cathelicid-OA1, from *Odorrana andersonii*, accelerates keratinocyte proliferation while demonstrating strong activity against Gram-negative bacteria [126]. These dual-functional peptides are especially valuable in managing chronic wounds and diabetic ulcers.

### 3.4. Current Biomedical Research and Future Applications

In recent years, the biomedical potential of amphibians has captured growing attention across pharmaceutical, medical, and ecological research fields. With increasing antimicrobial resistance, rising cancer prevalence, and a growing need for alternative treatments for metabolic and inflammatory diseases, amphibian-derived compounds offer a powerful and largely untapped resource for drug discovery. Peptides such as magainins, dermaseptins, brevinins, esculentins, and cathelicidins exhibit broad-spectrum antimicrobial activity and have shown efficacy against multidrug-resistant strains [183,184]. Modified analogs, such as pexiganan (a synthetic derivative of magainin II), have advanced to phase III clinical trials for diabetic foot ulcers [185]. Likewise, synthetic variants of dermaseptins and esculentin-1a undergo structural optimization to enhance selectivity and serum stability while minimizing cytotoxicity [186]. Novel hybrid peptides designed by combining domains from multiple amphibian peptides are being tested for synergistic effects against pathogens and biofilms [184]. Some peptides also exhibit dual roles combining antidiabetic and antimicrobial activity allowing for more integrated therapies [187]. Wound healing and antioxidant research is expanding due to the discovery of peptides like OA-GL21 and andersonin-G1, which accelerate keratinocyte migration, modulate redox responses, and stimulate TGF-β1 and GSH production [188]. Their ability to reduce oxidative stress while promoting tissue regeneration is now being studied in diabetic and burn wound models [189,190].

## 4. Amphibians as Bioindicators and Environmental Health Monitors

### 4.1. Sensitivity to Pollutants and Habitat Degradation

Amphibians are widely recognized as sentinels of environmental change due to their acute sensitivity to pollutants and habitat disturbances. Their physiology and complex life histories expose them to pollutants across habitats throughout their lives, resulting in the bioaccumulation of toxins in tissues, providing an integrated measure of contaminant exposure across ecosystems [191,192]. Contaminants from water, soil, and even air can accumulate in their tissues over time, relative to other taxa at similar trophic positions, making amphibians valuable indicators of environmental pollution [193].

Amphibians, particularly stream- and forest-dwelling salamanders, possess life-history, ecological, and physiological traits that make them highly effective indicators of environmental quality across diverse habitats. Jung et al. [194] developed a Stream Salamander Index of Biotic Integrity (SS-IBI) for small headwater streams in Maryland, USA integrating four metrics (species richness and composition, total salamander abundance, number of intolerant taxa, and number of adults) to capture biological responses to environmental stressors [194]. The SS-IBI correlated positively with forest cover and negatively with watershed impervious surface and reflected responses to acid-neutralizing capacity and dissolved organic carbon. Importantly, it remained effective even in areas of low salamander diversity, underscoring its utility where fish-based IBIs are impractical.

Stream-dwelling amphibians such as tailed frogs (*Ascaphus truei*), Pacific giant salamanders (*Dicamptodon tenebrosus*), and southern torrent salamanders (*Rhyacotriton variegatus*) are especially sensitive to habitat degradation from fine sedimentation and elevated water temperatures due to their dependence on interstitial spaces in coarse substrates for cover and foraging [195,196]. Amphibians serve as more consistent indicators than fish or macroinvertebrates due to a suite of key traits: longevity, high philopatry, stable populations, and tractability for monitoring [197,198]. Documented declines in the aforementioned species in the Pacific Northwestern US have been linked to logging activities, which reduce habitat quality and availability, highlighting their value as holistic, early-warning indicators of ecosystem stress [199,200].

The sensitivity of amphibians in montane and cloud forests to shifts in temperature and moisture makes them early indicators of climate-driven ecosystem change [201,202]. Similarly, the high abundance, site fidelity, small territory size, low fecundity, and sensitivity to microclimatic and successional changes exhibited by many plethodontid salamanders allows shifts in their numbers to reflect habitat quality [203]. Further, their low population variability compared to other vertebrates provides high statistical power for detecting ecological change, making them reliable sentinels for identifying habitat degradation [27,67].

These interactions between amphibians and their environment reveal complex ecological feedback. Amphibian sensitivity is species-specific, with some taxa showing high tolerance while others experience rapid declines under similar conditions [204]. This variability depicts the importance of community-wide monitoring while documenting species-level responses in ecological assessments. Given their unique ecological roles and physiological susceptibilities, amphibians are increasingly used in environmental risk assessments and biomonitoring programs [205].

### 4.2. Bioaccumulation of Heavy Metals and Pesticides

Amphibians are among the most effective vertebrate models for assessing the bioaccumulation of environmental contaminants, especially heavy metals and persistent organic pollutants as they readily absorb pollutants from the environment [206,207]. This bioaccumulation not only compromises individual fitness and population viability but also reflects broader ecological risks within affected ecosystems. Heavy metals (e.g., mercury, cadmium, lead, and arsenic) have been reported to interfere with metabolic pathways, disrupt endocrine signaling, and cause genotoxicity and immunosuppression in both larval and adult amphibians [176,208]. Amphibians exposed to pesticide-contaminated water often exhibit increased mortality, delayed metamorphosis, behavioral and developmental abnormalities, and endocrine disruption [207,209]. Bioaccumulation in amphibians has cascading ecological consequences as contaminants accumulated in amphibian tissues can be magnified to higher trophic levels, affecting ecosystem health and function [68]. Further, the susceptibility of amphibians to bioaccumulate a wide array of toxicants has led to their inclusion in global environmental monitoring programs. For instance, amphibians have been employed as bioindicators by governmental regulatory agencies in the USA, Europe and Asia [206,207]. These applications demonstrate the practical utility of amphibians in detecting early signs of ecological imbalance. Continued monitoring of amphibian bioaccumulation patterns and assessments of sublethal effects can provide valuable insight into emerging contaminant threats and support targeted mitigation efforts.

### 4.3. The Role of Amphibians in Metabolomics and Early-Warning Systems

Among the ecosystem services provided, amphibians offer unique insights into environmental metabolomics, a powerful tool that leverages small-molecule metabolic profiles to assess biological responses to pollutants. Environmental metabolomics involves analyzing biological samples, often using mass spectrometry or nuclear magnetic resonance spectroscopy, to detect and quantify metabolites [210]. Amphibians exposed to contaminants such as pesticides, herbicides, or heavy metals exhibit altered metabolic fingerprints that can be used to identify pollutant exposure and consequent physiological outcomes. For instance, Van Meter et al. [211] demonstrated how exposure to pesticide mixtures significantly altered the metabolomic profiles of post-metamorphic green frogs (*Lithobates clamitans*), providing early signals of environmental stress [211]. The sensitivity of this approach is exemplified by the detection of malaoxon, a malathion metabolite in frog tissues at low concentrations (0.68 ppm) using gas chromatography and mass spectrometry, enabling trace-level monitoring of organophosphate contamination in the field [211]. Metabolomic monitoring allows researchers to detect sublethal effects, chronic stress responses, and potential reproductive impairments long before visible signs of population decline [212]. Amphibian metabolite profiles thus function as environmental “canaries in the coal mine,” alerting scientists and policy makers to toxicological threats while there is room for mitigation. Miller et al. [213] argued for the broader application of environmental metabolomics in conservation, noting its ability to detect synergistic and cumulative effects of multiple stressors that elude conventional analytical techniques [211].

### 4.4. Indicators of Bioclimatic or Environmental Stress

Amphibians serve as sensitive indicators of bioclimatic stress and environmental perturbation and are among the first vertebrates to exhibit responses to climatic and environmental fluctuations, making them barometers for ecosystem change. Amphibians exhibit well-documented responses to climate change, such as range shifts, population declines, and altered phenology. For example, Parmesan [214] and Blaustein et al. [215] reported altitudinal and latitudinal migrations of amphibian populations in response to rising global temperatures [213,215]. These shifts often precede similar trends in other taxa, offering early warnings of broader ecosystem transformations.

Long-term studies in Europe and North America reveal that many amphibian species are now breeding earlier than historical norms, reflecting shifts in climate regimes [216,217,218]. In the montane and tropical cloud forest ecosystems, amphibians are especially vulnerable to climatic anomalies. The golden toad (*Incilius periglenes*) of Costa Rica’s Monteverde region is often cited as a case study of climate-induced extinction [219]. Despite being in a protected reserve, the species vanished in the late 1980s following a series of unusually dry seasons [220]. Besides climate, amphibians reflect other forms of environmental stress such as altered UV-B radiation levels, atmospheric pollution, ozone depletion, ecosystem acidification [209,221]. Therefore, amphibians offer a powerful lens through which to detect, understand, and anticipate environmental and bioclimatic stress. Their physiological plasticity, ecological diversity, and sensitivity to fine-scale environmental gradients position them as ideal indicator species in global change biology. Continued monitoring of amphibian populations not only informs conservation strategies for these taxa but also provides early detection systems for greater ecosystem disturbances.

## 5. Challenges and Knowledge Gaps in Amphibian Conservation

Amphibian species are currently facing an unprecedented global extinction crisis, with approximately 41% of species threatened with extinction, making them the most imperiled vertebrate class [27]. Their decline is driven by a convergence of anthropogenic and biological pressures, yet conservation interventions remain inadequate.

### 5.1. Habitat Destruction and Fragmentation

Habitat loss and degradation are widely recognized as the leading causes of amphibian declines worldwide [1]. This decline is primarily driven by anthropogenic activities. Commercial-scale agriculture is the single largest contributor, followed by timber harvesting and infrastructure development [222,223]. These activities result in the conversion, fragmentation, and destruction of core terrestrial habitats, aquatic breeding environments, and critical migration corridors, thereby reducing habitat complexity and diminishing ecological function [224,225]. Fragmentation reduces population sizes, increases isolation, and impairs dispersal, leading to heightened extinction risk [222,226]. Given the relatively small size, low vagility, and limited mobility (with dispersal ability often reported as <1 km) of many amphibian species, fragmentation critically compromises population connectivity, thus negatively affecting juvenile dispersal and reducing gene exchange, which undermines the long-term persistence and viability of local populations across degraded landscapes [223]. An additional ecological outcome of landscape-level land-cover change and fragmentation is biotic homogenization, observed as amphibian assemblages shift from diverse communities dominated by niche specialists to simplified communities primarily composed of generalist species, leading to a concurrent loss of both taxonomic and functional diversity [227].

### 5.2. Infectious Diseases and Pathogens

The decline in amphibian populations is often exacerbated by infectious diseases, which frequently interact synergistically with habitat loss and environmental stress [228]. The emergence of the fungal panzootic chytridiomycosis, caused by *Batrachochytrium dendrobatidis* (Bd), has been a major driver of global biodiversity loss, causing rapid declines and extinctions even in intact, protected habitats [1]. Furthermore, the introduction of pathogens, often facilitated by international trade and movement of amphibians for commerce, poses a risk of “pathogen pollution” to naïve wild populations [1,228]. A newly emergent, highly divergent fungal pathogen, *Batrachochytrium salamandrivorans* (Bsal), which primarily targets salamanders in the Western Palearctic, presents a severe future biosecurity threat [229]. Beyond chytrids, ranaviruses (Iridoviridae) also cause mass mortality events across multiple continents and can persist environmentally in aquatic systems for extended periods [230].

### 5.3. Climate Change

The influence of climate change is increasingly recognized not only as a critical driver of decline but also as an accelerating cofactor that compounds the effects of habitat loss and disease [1,223,231]. Amphibians, being ectotherms with highly permeable skin and reliance on specific thermal and hydric environments, are acutely sensitive to global warming [1]. The dominant climate-linked driver is drought, which disrupts critical processes like reproduction, development, and survival due to changes in water availability [1]. Other effects include habitat shifting and alteration and temperature extremes [1]. Climate change, including rising temperatures and altered precipitation patterns, compels species to make dynamic adjustments in their geographical distributions [223,231]. However, species with narrow distributional ranges or those coupled with specific habitats (e.g., exclusively forest or rupestrial environments) are projected to lose a greater portion of their suitable areas, as climate shifts exceed their limited dispersal capacity, increasing the risk of local extinction [231].

### 5.4. Lack of Local Engagement and Sociocultural Awareness

A critical but underappreciated barrier to amphibian conservation is the lack of community awareness and engagement. Despite the growing recognition of community-based conservation models, few amphibian-focused programs have successfully integrated indigenous knowledge or local stewardship. Yet, communities living near amphibian-rich habitats can play a vital role in habitat monitoring, data collection, and threat mitigation if properly empowered and trained [232]. Conservation efforts are also hindered by significant gaps in knowledge and policy. Invasive species, including bullfrogs and introduced fish, act as competitors, predators, and disease reservoirs, disproportionately affecting fragmented ecosystems [233]. Despite the scale of the crisis, amphibians also receive disproportionately low conservation funding compared to other vertebrates, a reflection of taxonomic bias and weak institutional coordination [234]. Over 25% of amphibian species remain classified as Data Deficient (IUCN, 2023; https://www.iucnredlist.org; accessed on 30 December 2025), and long-term monitoring remains rare, limiting detection of population trends or emerging threats. Sociocultural stigma and low community engagement further constrain local stewardship. Yet, integrating ecological, immunogenetic, and disease-monitoring frameworks with inclusive, community-based conservation strategies will be essential to mitigate future declines and protect amphibian biodiversity [235].

## 6. Conservation Strategies

Amphibian conservation must be grounded in recognition of their ecological, biomedical, economic, and sociocultural value. Amphibians perform essential ecosystem functions that link terrestrial and aquatic systems, yet these roles are rarely accounted for in conservation policy or land management decisions [236]. Incorporating the ecosystem service value of amphibians into environmental planning could incentivize habitat protection beyond species-focused approaches. For example, frog-friendly agricultural practices support amphibian populations while also generating marketable eco-certification for farmers [237]. Additionally, amphibians are a critical reservoir of bioactive compounds, underscoring their importance to global health innovation [238,239].

Despite their importance and contributions, amphibians remain underfunded and undervalued in conservation agendas [240]. They receive less than 1% of global biodiversity funding, are poorly represented in national action plans, and are frequently overlooked in land use and public health frameworks [37,241]. Conservation must move beyond reactive species-based interventions and adopt integrative strategies that preserve genetic, ecological, and functional diversity while addressing broader threats such as habitat loss, climate change, and disease. This includes supporting genetic monitoring to maintain immunogenetic resilience, investing in wetland restoration and nature-based solutions, and promoting community-based stewardship and environmental education (Table 3) [242]. Captive colonies of rare and threatened species can serve as critical safeguards against extinction. These captive populations can support ex situ conservation by maintaining viable populations for potential reintroduction [222]. Bridging science with economic incentives, health research, and sociocultural engagement is essential to elevate amphibians within global conservation priorities and ensure the long-term stability of the ecosystems they support [243].

## 7. Future Directions and Research Gaps

Despite a growing body of literature emphasizing the ecological, biomedical, and socio-economic importance of amphibians, significant gaps remain that warrant urgent scientific attention. Most existing studies concentrate on a few well-studied amphibian species, particularly from North America and parts of Europe, leaving vast regions such as sub-Saharan Africa, Southeast Asia, and the Neotropics underrepresented in conservation and ecological impact analyses [263,264]. There is a pressing need to extend biogeographic and functional assessments of amphibian roles in nutrient cycling, pest control, and ecosystem engineering to a broader range of habitats and taxa (Figure 2).

In biomedical research, while AMPs like magainins and dermaseptins have shown promising pharmacological properties, very few have progressed to clinical trials. Furthermore, the biochemical mechanisms underlying regenerative capabilities in salamanders, particularly their potential applicability to mammalian systems, remain incompletely understood. Future research should prioritize high-throughput screening of amphibian-derived bioactive compounds and investigate their synergy with existing pharmacological agents to overcome antimicrobial resistance. Despite their ecological significance, the socio-economic contributions of amphibians to rural livelihoods, traditional medicine, and ecotourism, remain poorly quantified. There is a pressing need to standardize valuation frameworks that account for both market and non-market ecosystem services provided by amphibians, enabling their fuller integration into conservation planning and policy. Additionally, amphibians’ role as early-warning indicators in environmental metabolomics and endocrine disruption science is still emerging, with untapped potential for integrating transcriptomics and proteomics to assess ecosystem health. Advancing amphibian conservation will require interdisciplinary approaches that integrate remote sensing, citizen science, and participatory conservation frameworks. These tools are especially valuable for monitoring cryptic or elusive species and for enabling real-time assessment of habitat degradation and conservation effectiveness. There is a critical need for robust policy frameworks that explicitly recognize the multifunctional value of amphibians across biodiversity conservation, human health, and sustainable development.

Indigenous and traditional ecological knowledge (TEK) remains underexplored in amphibian research and conservation [265,266]. TEK can provide valuable insights into amphibian ecology, phenology, and habitat use particularly in remote regions where scientific data are limited [267]. TEK offers culturally grounded perspectives on the significance of amphibians, which are often embedded in local cosmologies, resource practices, or taboos. Incorporating TEK into future conservation frameworks would not only enrich ecological understanding but also enhance the cultural relevance and social legitimacy of management strategies. As such, integrating TEK should be prioritized as a future direction to foster inclusive, place-based conservation and to better capture the multidimensional value of amphibians.

A promising future direction lies in leveraging amphibians as focal species for public engagement and environmental education (Figure 2). Their sensitivity to environmental change, local visibility, and unique life histories make them compelling subjects for building ecological awareness and fostering community stewardship. However, amphibians are often culturally stigmatized, particularly in rural and marginalized communities, which limits conservation support. Targeted outreach that integrates culturally relevant narratives, TEK, and participatory learning such as school-based programs, citizen science, and community storytelling can help shift perceptions and build localized conservation ethics [268,269].

A critical future direction also lies in the systematic integration of emerging technologies to quantify and generalize amphibian environment relationships across biomes. While amphibians are widely regarded as sentinels of environmental change, this status has often been based on local or qualitative observations. To elevate their role as robust indicators, technologies such as remote sensing, geographic information systems, and automated biomonitoring must be leveraged to detect habitat change, degradation, and climate-driven stress at ecologically relevant scales [270]. High-resolution satellite imagery, spectral indices such as normalized difference vegetation index (NDVI), enhanced vegetation index (EVI), and surface moisture indices can identify shifts in wetland and vegetation health parameters closely linked to amphibian habitat suitability [271].

Additionally, the expansion of citizen science and sensor-based monitoring platforms presents an opportunity to generate large-scale time series datasets that capture phenological and demographic responses to environmental stressors [272]. When paired with standardized statistical frameworks and machine learning approaches, these data can be used to derive predictive models that quantify amphibian responses to land-use and climate gradients across regions. Such efforts would enhance the empirical basis for using amphibians as early-warning indicators and strengthen their utility in environmental monitoring programs globally. Establishing this generalized, quantifiable framework is essential for translating amphibian-based bioindication into policy-relevant metrics that inform conservation, land management, and climate adaptation planning.

## 8. Conclusions

Amphibians occupy a critical ecological and functional niche across a range of ecosystems. Yet, despite these multifaceted roles, amphibians are among the most threatened vertebrate taxa on Earth. This alarming decline underscores not only a conservation crisis but also the erosion of essential ecosystem functions and services they underpin. This review has comprehensively synthesized global knowledge on amphibians from ecological, biomedical, and socio-economic perspectives. Furthermore, their sensitivity to environmental changes ranging from chemical contaminants to climate-induced stressors makes them bioindicators for ecosystem health monitoring. Beyond their ecological services, they contribute to livelihoods through traditional medicine, cultural symbolism, and bioprospecting, especially in biodiversity-rich regions. Amphibians also hold potential for sustainable ecotourism, human sustenance, education, and community-based conservation, offering economic incentives aligned with biodiversity protection.

Despite the wealth of scientific insight and ecological importance, amphibians remain grossly underrepresented in research funding and conservation planning. The transdisciplinary importance of amphibians necessitates urgent integration of ecological science and socio-economic strategy. Protecting amphibians is not merely an act of preserving biodiversity; it is an investment in the resilience of ecosystems, the advancement of medical science, and the sustainability of human well-being. Future initiatives must prioritize large-scale habitat protection, the mitigation of emerging infectious diseases such as chytridiomycosis, and the enforcement of regulations concerning pollution and invasive species. Expanding research into the molecular and physiological mechanisms behind amphibian traits, especially their immunological defenses and regenerative processes can open new frontiers in medicine and biotechnology. Monitoring amphibian populations for environmental stress, supported by metabolomics and remote sensing tools, may also enhance global responses to climate change and pollution. A renewed global commitment to amphibian research and conservation will not only protect these vital organisms but will also fortify the ecosystems and societies that depend on them.

## Figures and Tables

**Figure 1 biology-15-00098-f001:**
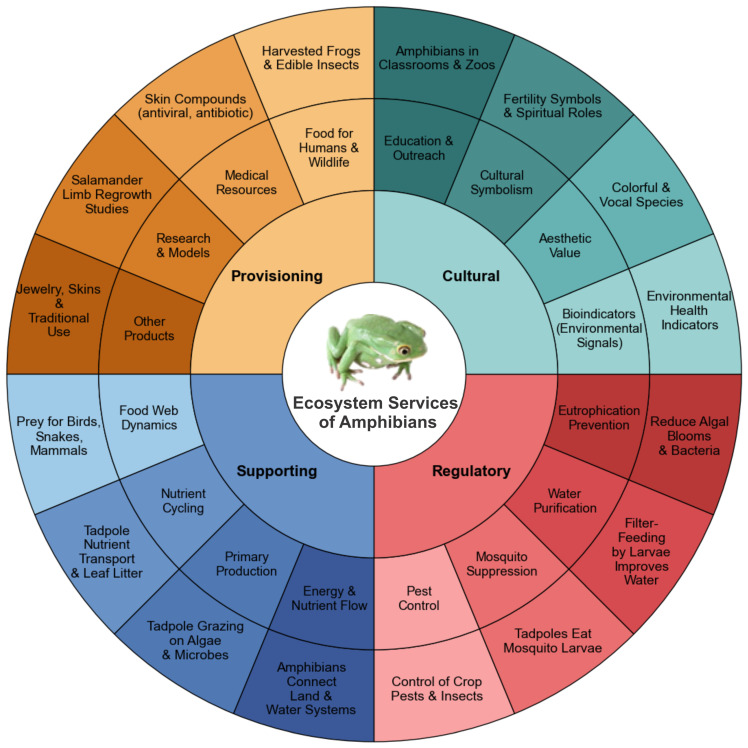
Ecosystem functions provided by amphibians. This figure exhibits the main four ecosystem services rendered by amphibians including the provisioning, supporting, regulatory and cultural services (source: created by authors).

**Figure 2 biology-15-00098-f002:**
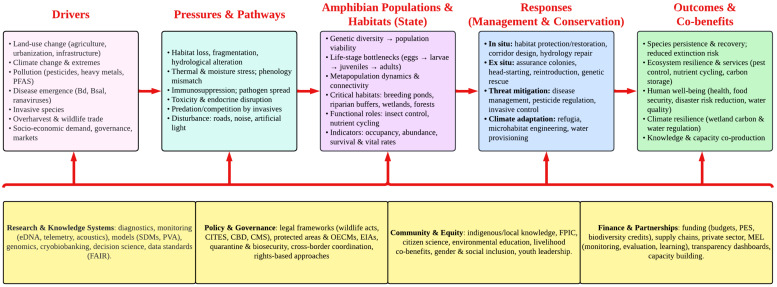
A conceptual framework for amphibian conservation integrating socio-ecological drivers, ecological responses, and cross-sectoral solutions outlining the pathway from global drivers to impacts on amphibian populations and habitats, and corresponding management and conservation responses.

**Table 2 biology-15-00098-t002:** Major bioactive compounds in amphibians and their medical applications.

Main Compound	Compound Name	Amphibian Species	Origin	Medical Use	Mechanism/Activity	Reference
Alkaloid	Zetekitoxin-AB	*Atelopus zeteki*	Panama	Antibacterial	MIC 10 µg/mL against *E. coli*	[105]
	Bufotenine	*Bufo gargarizans*	China	Antitumor	Inhibits cell proliferation and induces apoptosis	[131]
	Pyrrolizidine	*Melanophryniscus* spp.	Argentina, Brazil, Paraguay, Uruguay	Antibacterial	MIC 3.9 µg/mL against *S. aureus*, and 31.3 µg/mL against *E. coli*	[132]
Peptide	Bombesin-LS	*Bombina orientalis*	Korea	Hormonal	Stimulates release of gastrin	[133]
	RP-Bombesin	*Boana raniceps*	Argentina	Antibacterial, Hormonal	MIC 25 µg/mL (*S. aureus*), stimulates gastrin	[106]
	Brevinin-1	*Lithobates pipiens*	Canada	Antibacterial, Antidiabetic	MIC 26 µg/mL (*E. coli*), stimulates insulin	[126]
	Brevinin-2GU	*Hylarana guntheri*	China	Immunomodulatory	Reduces TNF-α, suppresses IFN-γ	[111]
	Magainin-1	*Xenopus laevis*	Africa	Antibacterial	MIC 5 µg/mL (*E. coli*), 50 µg/mL (*S. aureus*)	[120]
	Magainin A	*Bombina maxima*	China	Anticancer, Antiviral	Inhibits tumor growth, antiviral properties	[112,113]
	Cruzioseptin-16	*Cruziohyla calcarifer*	Costa Rica	Antibacterial	Resistant to multiple bacteria	[114]
	Esculentin-1a	*Lithobates areolatus*	USA	Antibacterial, Wound Healing	Stimulates keratinocyte migration	[115]
	Ocellatin-4	*Leptodactylus* spp.	South America	Antibacterial	MIC 64 µM (*E. coli*, *S. aureus*)	[116]
	Proline-arginine (RP-27)	*Rhinophrynus dorsalis*	Mexico	Antidiabetic	Stimulates insulin secretion	[107]
	Syphaxin	*Leptodactylus syphax*	Brazil	Antibacterial	Low MICs against bacteria	[108]
	Andersonin-C1	*Odorrana margaretae*	China	Antioxidant	Scavenges ROS via amphipathic helix formation	[101]
	Andersonin-G1	*Odorrana andersonii*	China	Antioxidant	Inhibits lipid peroxidation and chelates metals	[101]
	Andersonin-H3	*Odorrana margaretae*	China	Antioxidant	Radical scavenging, sequence	[101]
	Antioxidin-RP1	*Rana pleuraden*	China	Antioxidant	Scavenges ABTS, DPPH radicals	[109]
	APBSP	*Lithobates catesbeianus*	USA	Antioxidant	Quenches DPPH, hydroxyl, superoxide radicals	[130]
	Cathelicid-OA1	*Odorrana andersonii*	China	Antioxidant, Wound healing	Boosts catalase, GSH; accelerates skin healing	[124]
	Nigroain-B-MS1	*Hylarana maosuoensis*	China	Antioxidant	High ABTS and DPPH scavenging	[110]
	OA-VII2	*Odorrana andersonii*	China	Antioxidant	Enhances CAT, SOD in vivo	[101,134]
	OA-GL21	*Odorrana andersonii*	China	Antioxidant	ABTS scavenging activity	[101]
	Pleurain-A1	*Rana pleuraden*	China	Antioxidant	ABTS, NO scavenging	[101,109]
	Buforin-2	*Bufo bufo gargarizans*	Unknown	Antimicrobial	Broad spectrum activity incl. fungi	[135]
	Dermaseptin-S1	*Phyllomedusa sauvagii*	South America	Antimicrobial	Binds and disrupts membranes	[136]
	Esculentin-1	*Rana esculenta*	Europe	Antimicrobial, Antidiabetic	Broad MIC, insulin-stimulating	[127]
	Hymenochirin-1B	*Hymenochirus boettgeri*	Africa	Antidiabetic, Anticancer	Insulin release and tumor inhibition	[127,137]
	Dermaseptin-PP	*Phyllomedusa palliata*	South America	Anticancer	Induces apoptosis in tumor cells	[138,139]
	XLAsp-P1	*Xenopus laevis*	Africa	Anticancer	Membrane disruption of cancer cells	[128]
	Phylloseptin-L2	*Hylomantis lemur*	Costa Rica	Antidiabetic	Stimulates insulin secretion	[117,127]
	Plasticin-L1	*Leptodactylus laticeps*	South America	Antidiabetic	Enhances insulin release	[118,127]
	Pseudhymenochirin-1Pb	*Pseudhymenochirus merlini*	Africa	Antidiabetic, Anticancer	Multiple bioactivities	[140,141]
	Alyteserin-2a	*Alytes obstetricans*	Europe	Antidiabetic	Increases insulin release	[142]
	Amolopin	*Amolops loloensis*	Unknown	Antidiabetic	Enhances insulin release in vitro	[127]
	Bombesin (protein fractions)	*Bombina variegata*	Europe	Antidiabetic	Stimulates insulin secretion	[127,129]
	Brevinin-1CBb	*Lithobates septentrionalis*	Canada	Antidiabetic	Increases insulin secretion, low cytotoxicity	[143]
	Caerulein-B1	*Xenopus borealis*	Unknown	Antidiabetic	In vitro insulinotropic peptide	[127,144]
	CPF-AM1	*Xenopus amieti*	Africa	Antidiabetic	Stimulates GLP-1 release	[145]
	Dermaseptin B4	*Phyllomedusa trinitatis*	South America	Antidiabetic	Unknown insulin-stimulating mechanism	[146,147]
	Esculentin-2Cha	*Lithobates chiricahuensis*	USA	Antidiabetic	GLP-1 activity, plasma insulin increase	[148]
	GM-14	*Bombina variegata*	Europe	Antidiabetic	Stimulates insulin secretion	[127,129]
	IN-21	*Bombina variegata*	Europe	Antidiabetic	Insulin-releasing bioactive peptide	[127,129]
	GPPGPA	*Andrias davidianus*	China	Antidiabetic	α-glucosidase inhibitor	[149]
	Magainin-AM1	*Xenopus amieti*	Nigieria, Africa	Antidiabetic	GLP-1 release, antimicrobial	[145,150]
	Ocellatin-L2	*Leptodactylus laticeps*	South America	Antidiabetic	Boosts insulin output	[118,151]
	Palustrin-2CBa	*Lithobates catesbeianus*	USA	Antidiabetic	Insulin secretion enhancer	[143]
	PGLa-AM1	*Xenopus amieti*	Africa	Antidiabetic	GLP-1 enhancement	[145,152]
	Ranateurin-2CBc	*Lithobates catesbeianus*	USA	Antidiabetic	Increases insulin release	[143]
	Temporin-DRa	*Rana draytonii*	USA	Antidiabetic	Low toxicity, insulinotropic	[153]
	Temporin-Oe	*Rana ornativentris*	Unknown	Antidiabetic	Moderate insulin stimulation	[153]
	Tigerinin-1R	*Rana tigerina*	Unknown	Antidiabetic	Regulates glucose metabolism	[145,154]
	Xenopsin	*Xenopus amieti*	Africa	Antidiabetic	Insulin-releasing short peptide	[145,152]
	AH90	*Odorrana grahami*	Unknown	Wound Healing	Stimulates TGF-β1 in healing	[155]
	Temporin-1RNb	*Rana nigromaculata*	East Asia	Antimicrobial	Inhibits multiple pathogens	[135,156]
	Nigrocin-1	*Rana nigromaculata*	East Asia	Antimicrobial	Effective against Gram-negative bacteria	[135]
	Nigrocin-2	*Rana nigromaculata*	East Asia	Antimicrobial	Broad antimicrobial spectrum	[135]
	Maximin-H5	*Bombina maxima*	Asian	Antimicrobial	Inhibits *S. aureus* growth	[106,157]
	Japonicin-2	*Rana chaochiaoensis*	China	Antimicrobial	Effective against *S. aureus* and *E. coli*	[158]
	Dermaseptin-B3	*Phyllomedusa bicolor*	South America	Anticancer	Inhibits PC-3 prostate tumor cells	[147,159]
	Temporin-1CEa	*Rana chensinensis*	China	Anticancer	Selective cytotoxicity in cancer cells	[160]
	Temporin-Vc	*Lithobates virgatipes*	USA	Antidiabetic	High insulin-releasing activity	[127,153]
	RK-13	*Agalychnis calcarifer*	Costa Rica	Antidiabetic	Short insulinotropic peptide	[161]

**Table 3 biology-15-00098-t003:** Summary of global conservation efforts and their effectiveness.

Region/Country	Conservation Strategy/Program	Target Species	Effectiveness/Outcome	Reference
Central America	Amphibian Ark/Captive Breeding	*Atelopus* spp., *Craugastor* spp.	reintroductions limited by habitat threats	[2,244]
Australia	Disease management (chytrid fungus)	*Litoria* spp., *Taudactylus* spp.	ex situ programs	[245]
Europe	Natura 2000/EU Habitats Directive	*Bombina bombina*, *Alytes* spp.	Enhanced protection of critical breeding habitats	[246,247]
USA	Headstart and release programs	*Cryptobranchus a. alleganiensis*	Viable strategy for rebuilding eastern hellbender populations	[248]
India	Community education & monitoring programs	*Nyctibatrachus* spp., *Indirana* spp.	Improved local awareness; limited enforcement capacity	[1,249]
Brazil	Protected area expansion (Amazon reserves)	*Leptodactylus* spp., *Dendrobates* spp.	Positive buffer against deforestation-related declines	[67,250]
Colombia	Red List and priority site mapping	Andean stream frogs	identify hotspots for focused conservation	[67,251]
South Africa	Biodiversity stewardship programs	*Breviceps* spp., *Hyperolius* spp.	Enhanced private land protection; requires long-term funding	[252]
Costa Rica	Environmental education + long-term monitoring	*Craugastor* spp., *Hyalinobatrachium* spp.	Improved data collection; increased local conservation action	[253]
France	Reintroduction of *Bombina variegata*	*Bombina variegata*	Successful site-level reestablishment; requires continuous monitoring	[254]
Ecuador	Yasuní National Park amphibian inventories	*Pristimantis* spp., *Dendrobatidae*	High biodiversity documentation; improved policy influence	[255,256]
China	Habitat protection and legislation (e.g., CITES enforcement)	*Rana chensinensis*, *Andrias davidianus*	Controlled trade and supported captive breeding	[257,258]
Italy	Genetic monitoring and relocation programs	*Salamandra atraaurorae*	Maintained genetic diversity in isolated populations	[259]
Japan	Wetland reserves for endemic frogs	*Glandirana rugosa*, *Buergeria* spp.	Enhanced breeding site protection	[260,261]
Thailand	Integrated conservation and development	*Kalophrynus* spp., *Leptolalax* spp.	Community acceptance increased with benefit-sharing	[262]

## Data Availability

No new data were created or analyzed in this study. Data sharing is not applicable to this article.

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
