# Peer review of "The Multifaceted Importance of Amphibians: Ecological, Biomedical, and Socio-Economic Perspectives"

_biology, 2026, doi:10.3390/biology15010098_

Round 1

Reviewer 1 Report (Previous Reviewer 1)

Comments and Suggestions for Authors

Your comprehensive revision has produced an excellent paper.

Reviewer 2 Report (Previous Reviewer 2)

Comments and Suggestions for Authors

The authors have addressed my comments appropriately and I have no further concerns. In my opinion, the manuscript is a comprehensive synthesis that would make a valuable contribution to the literature.

This manuscript is a resubmission of an earlier submission. The following is a list of the peer review reports and author responses from that submission.

Round 1

Reviewer 1 Report

Comments and Suggestions for Authors

The review has the potential to make a significant contribution to guide amphibian conservation and research. However, it is far too long due to many paragraphs that could be summarised in a sentence or two, referring to other reviews or research papers. The review focuses too much on field conservation. It does not refer to conservation breeding programs or biotechnologies in their increasing role and potential for amphibian conservation, or, for instance, amphibians' contribution to cloning and endocrinology. These should be included along with medicine in an expanded biotechnologies section. A general revision should make the review cohesive and balanced, as it reads as if different authors wrote different parts. Attached is a PDF with notes on some of these potentials. 

Comments on the Quality of English Language

The review reads as if different parts were written by different authors and is too verbose in parts. English expression could be improved throughout. 

Author Response

General comments:

Reviewer comment (RC): The review has the potential to make a significant contribution to guide amphibian conservation and research. However, it is far too long due to many paragraphs that could be summarised in a sentence or two, referring to other reviews or research papers. The review focuses too much on field conservation. It does not refer to conservation breeding programs or biotechnologies in their increasing role and potential for amphibian conservation, or, for instance, amphibians' contribution to cloning and endocrinology. These should be included along with medicine in an expanded biotechnologies section. A general revision should make the review cohesive and balanced, as it reads as if different authors wrote different parts. Attached is a PDF with notes on some of these potentials.

Author Response (AR): Thank you for this great feedback. We have cut down the text substantially (from a total of 46 pages to 39 pages in the revised version, the text portion of the ms is 24 pages with figures and tables), included more information on medical importance with respect to endocrinological (L366-370) aspects and multiple biomedical aspects expanding the biomedical section 3 (L 333-482), and reduced the focus on conservation. We also fixed all the typos and other edits as suggested.

That said, our article is a comprehensive review, not a systematic review. Therefore, the value of the article lies in detailing the context and the findings. Simply summarizing these details into a single sentence will not do justice to the articles we reviewed nor what is expected of a comprehensive review.  Following your excellent suggestions, we revised the entire abstract, throughout the body of the text, and the Conclusion.

Specific comments

RC: the word “Biomedical” in the title should be changed to “Biotechnical”.

AR: We changed title as suggested, now the title reads as "The Multifaceted Importance of Amphibians: Ecological, Biotechnical, and Socio-Economic Perspectives".

RC: this theme is common in recent literature

AR: Yes, we agree. That is precisely why we are reviewing this topic.

RC: Permeable physiology, what does this mean?

AR: This refers to their highly permeable skin, we changed the wording in the abstract

RC: Bioindicators in-place of sentinel species

AR: We applied this change throughout the ms as suggested

RC:  this is not a paradigm shift  -use 'greater integration'

AR: We agree and changed the text into “increased integration”

RC: traditional instead of indigenous

AR: we changed this wording

RC: replace “nature” with “the wild”

AR: we applied this change

RC: “class” in place of “clade”

AR: we applied this change

RC: “described” in place of “recognized”

AR: we applied this change

RC: “terrestrial and freshwater” in place of “non-marine”

AR: we applied this change

RC: there have been declines of birds and reptiles recorded in pristine habitats

AR: we agree, but, our statement is a general remark that is widely agreed up as supported by citations, this is not not an absolute statement.

RC: on section 1.3: This section is summarised into the previous sentence with a reference to official IUCN reports and some specific articles

AR: we removed the repetition in these two paragraphs

RC:  “increased” in place “of proliferation”

AR: We applied this change

RC: “reduced snake” in place of “snake decline”

AR: we applied this change

RC: section 1.4. This is better if more summarised and moved to the last paragraph in his section.

AR: great suggestion, edited as suggested

RC: changing the figure caption for Figure 1.

AR: We revised the caption to: “Figure 1: Ecosystem services of amphibians. This figure exhibits the main four ecosystem services rendered by amphibians including the provisioning, supporting, regulatory and cultural services”

RC: as the species are not mentioned above the Dominant Am .. column can be removed. Use abbreviations for terms ie. Not reported as NR and notes in table tile.

AR: Agreed, and this column is now removed

RC: that along with sequestration into larval amphibians skeletons

AR: we included this part into the text

RC: “composition and concentration” in place of “stoichiometry”.

AR: we applied this change

RC: “a process accelerated in shallow ephemeral environments”

AR: we applied this change

RC: “post metamorphosis becoming terrestrial”

AR: we applied this change

RC: under section 2.2: better to have the average number of eggs and possibly their yolk diameter

AR: this is an excellent suggestion, but, unfortunately, this reference only includes the per sqm density not an average. The publication did not report the yolk diameter either 

RC: A paragraph about captive assurance colonies along with reproduction and advanced biotechnologies is needed to save species whose habitat no longer exists.

AR: we included this point into the ms (L 689-694). We kept this section short and to the point.

RC: Incorporating captive assurance colonies and biotechnologies into the figure.

AR: This is an excellent suggestion. We revised the entire figure 2 to include these points

RC: In the conclusion: the are thermoconformers or behavioural thermoregulators - 'add " as vertebrates" as there are plenty  of invertebrates

AR: We substantially cut down the conclusion section and this reference was removed entirely to avoid repetition.

RC: in the conclusion: policy and conservation planning seem like the same thing  

AR: yes, good point, these are related. We removed “policy” reference

RC: is biomedical innovation  dependent on ecological science?

AR: good point, we reframed this sentence for clarity as well

RC: In reference to Table 1, suggested that the "Dominant Amphibian" column could be removed since the species were not mentioned above the table, and recommended using the abbreviation NR (for Not reported) in the table and noting this abbreviation in the table title.

AR: we deleted the “Dominant Amphibian” column, replacing "Not reported" with **NR**, and noting this change in the table title.

RC: Regarding lines L 264 - 271, add the phrase “that along with sequestration into larval amphibians skeletons”.

AR: We have edited this part as suggested.

RC: In line L 765, the reviewer suggested deleting the word “ecological”.

AR: we deleted “ecological”, and the revised text reads like: “Bridging science with economic incentives, health research, and sociocultural engagement is essential to elevate amphibians within global conservation priorities and ensure the long-term stability of the ecosystems they support ”.

A summary of line by line comments from the PDF

No

Line Number/ Section

Comments/Address (New Line Number)

1

Title

“Biomedical” changed to Biotechnical

Revised

The Multifaceted Importance of Amphibians: Ecological, Biotechnical, and Socio-Economic Perspectives

2

Simple Summary

Revised

Amphibians—frogs, toads, salamanders, and caecilians—are disappearing at an alarming rate around the world. Scientists estimate that over 40% of species are now threatened with extinction. This review explains why amphibians are so important—not just for biodiversity, but also for healthy ecosystems and human well-being. Amphibians help control pest insects, recycle nutrients, and provide food for many other species. They also have medical value because their skin contains compounds that may lead to new medicines. Unfortunately, habitat destruction, climate change, pollution, invasive species, and diseases are driving many populations to decline. This paper gathers current knowledge about the roles amphibians play in the wild and explains how their decline can disrupt ecosystems. It also highlights how new technologies—like satellite mapping and citizen science—can help track these changes and support conservation efforts. Importantly, the review calls for better public education, use of Indigenous knowledge, and stronger policies that recognize amphibians not just as endangered species but as vital parts of functioning ecosystems. Protecting them is not only about saving frogs—it’s also about protecting nature’s balance and our own health

3

Abstract

Revised

4

L 78 - 101

Revised

5

L 106 - 110

Revised

6

L 146 - 161

Revised

7

L 162 - 182

Revised

8

L 195 - 216

Revised

This review provides an integrated and comprehensive understanding of the multifaceted importance of amphibians by synthesizing current research across ecological, biotechnical, and socio-economic domains. We synthesize peer-reviewed literature, international reports, and conservation datasets published primarily between 2000 and 2024, with in-depth analyses on studies from the last five years. Sources were identified using structured keyword searches in academic databases such as Web of Science, PubMed, and Scopus. Key search terms included amphibia in combination with “ecosystem services,” “ecosystem functions,” 9“ecological processes,” “ecological role,” “ecological importance,” and “bioindicators.” The selected literature was analyzed qualitatively to identify recurring themes, novel findings, conservation implications, and knowledge gaps. By contextualizing these dimensions within the global amphibian decline, we underscore the urgent need for interdisciplinary conservation strategies and offer a framework for integrating amphibians into broader sustainability, health, and biodiversity agendas.

9

Figure 1

Change the figure caption

Revised

New figure caption

Figure 1: Ecosystem services of amphibians. This figure exhibits the main four ecosystem services rendered by amphibians including the provisioning, supporting, regulatory and cultural services (source: created by authors).

10

L 242 - 261

Revised

Tadpoles play a foundational role in many freshwater ecosystems, particularly in humid tropical regions such as the Amazon and Southeast Asia. Their exceptionally high densities in some ecosystems allow them to exert strong influences on benthic structure, primary production, and nutrient fluxes.

Through intensive grazing and sediment-feeding, tadpoles regulate algal and periphyton communities and shift assemblages toward grazing-resistant taxa. Simultaneously, they modulate sediment dynamics via bioturbation and particle ingestion, decreasing total benthic sediment loads, including organic and inorganic fractions, by 41–43%.

These processes increase nutrient availability and facilitate higher densities of other consumers. Such sediment-feeding activities can also create competition with sediment-feeding fish while facilitating others that rely on exposed aquatic substrates.

Additionally, mass metamorphic emergence in both tropical and temperate systems transfers substantial biomass up to 2,500 kg/ha/year from aquatic to terrestrial ecosystems, functioning as a spatial subsidy that reinforces amphibians’ foundational role across ecosystem boundaries.

11

Table 1

as the species are not mentioned above the Dominant Amphibian column can be removed. Use abbreviations for terms ie. Not reported as NR and notes in table tile.

Deleted the Dominant Amphibian column

Replaced, Not reported as NR and noted in the table tile

12

L 264 - 271

Add “that along with sequestration into larval amphibians skeletons”

Revised

13

L 272 - 284

Revised

14

L 751

Revised the whole sentence

For example, tree frog-friendly agricultural practices support amphibian populations while also generating marketable eco-certification for farmers.

15

L 765

Deleted “ecological”

Deleted

Bridging science with economic incentives, health research, and sociocultural engagement is essential to elevate amphibians within global conservation priorities and ensure the long-term stability of the ecosystems they support.

16

Conclusion

L 843 - 883

Revised

Reviewer 2 Report

Comments and Suggestions for Authors

Summary

The aim of this review was to gather multidisciplinary evidence of the importance of amphibians by integrating current understanding of their ecological, socio-economic and biomedical value. In doing so, the authors highlight the underappreciated significance of amphibians for ecosystem functioning and human wellbeing. The authors use this multidisciplinary approach to argue for a paradigm shift whereby amphibian conservation is leveraged by broader frameworks including ecosystem service valuation, climate resilience planning and public health policy. The review concludes by identifying current conservation challenges, knowledge gaps and future research opportunities to help fast-track conservation of one of the most threatened vertebrate taxa globally.

In my opinion, this manuscript makes a significant and worthwhile contribution to the literature by assimilating and integrating the most up to date literature on the importance of amphibians across disciplines, providing a unique perspective on their value to ecosystem functioning and human health that to date has been somewhat underappreciated, especially in policy. This topic is highly relevant given the urgency of amphibian conservation in the face of ever-increasing threats. The review is well written with a clear structure and provides a comprehensive coverage of the relevant literature, which is also summarised into figures and tables to aid the reader. I have only minor comments (see below), and with appropriate revisions made I believe this review will make an important publication.

General comments

The review is very comprehensive. However, I feel that it falls a bit short on two subject matters that are raised in the introduction as being relevant to the review. These are (1) the role of amphibians in eco-tourism, and (2) amphibians as a food source for humans (both have cultural/socio-economic elements). Are there any relevant examples on these topics that the authors can integrate into the text? Even a few extra sentences would help bolster the review’s coverage of the socio-economic importance of amphibians.

Specific comments

L167: Please define “poikilotherm” and provide some context as to how this life adaptation/strategy enables amphibians to accumulate high biomass relative to energy input.

Figure 1: this figure should be referred to in-text.

L242: Given that tadpoles are a life history stage, not a species, this sentence may be better re-phrased as something along the lines of “Tadpoles play a foundational role in many freshwater ecosystems…”

L292: Please provide the full scientific name and italicise P. serratus.

L329-332: be careful switching between present and past tense. This sentence would be better written in present tense.

Table 2: Check that all scientific names are italicised throughout all columns.

L443: Ideally, Table 2 should be below its first mention in the text. Same for Figure 2.

L520: Please italicise “laevis”.

L751: Author citation missing.

Table 3: Please italicise “Bombina variegata”

Table 3: No ecosystems are listed in column 3, so I would suggest changing the header to just “Target Species”.

Figure 2: This figure is informative but has several spelling errors. Additionally, the title is unnecessary given that it is near-identical to the figure caption. I suggest deleting the overhead title.

References: There are species names throughout the Reference list that need to be italicised.

REF 23, L961: There is some incorrect spacing in this reference.

REF 34, L990: Author name needs capitalisation.

REF 86, L1117: delete space in “Appalachian”

REF 108, L1167: This reference has already been listed (reference 51) and should be deleted. Remember to update the relevant in-text citations appropriately.

REF 129, L1223: this is not referenced properly.  

REF 146, L1272: Please provide author details for this reference.

Author Response

General comments

RC: The aim of this review was to gather multidisciplinary evidence of the importance of amphibians by integrating current understanding of their ecological, socio-economic and biomedical value. In doing so, the authors highlight the underappreciated significance of amphibians for ecosystem functioning and human wellbeing. The authors use this multidisciplinary approach to argue for a paradigm shift whereby amphibian conservation is leveraged by broader frameworks including ecosystem service valuation, climate resilience planning and public health policy. The review concludes by identifying current conservation challenges, knowledge gaps and future research opportunities to help fast-track conservation of one of the most threatened vertebrate taxa globally.

In my opinion, this manuscript makes a significant and worthwhile contribution to the literature by assimilating and integrating the most up to date literature on the importance of amphibians across disciplines, providing a unique perspective on their value to ecosystem functioning and human health that to date has been somewhat underappreciated, especially in policy. This topic is highly relevant given the urgency of amphibian conservation in the face of ever-increasing threats. The review is well written with a clear structure and provides a comprehensive coverage of the relevant literature, which is also summarised into figures and tables to aid the reader. I have only minor comments (see below), and with appropriate revisions made I believe this review will make an important publication.

The review is very comprehensive. However, I feel that it falls a bit short on two subject matters that are raised in the introduction as being relevant to the review. These are (1) the role of amphibians in eco-tourism, and (2) amphibians as a food source for humans (both have cultural/socio-economic elements). Are there any relevant examples on these topics that the authors can integrate into the text? Even a few extra sentences would help bolster the review’s coverage of the socio-economic importance of amphibians.

AR: Thank you for this valuable feedback. As noted in the manuscript, we acknowledge the relevance of ecotourism and the food value of amphibians. These aspects are touched upon in our discussion of ecosystem services and livelihood linkages (lines 721, 785). A dedicated, in-depth analysis was omitted for two reasons: (1) to maintain a focused scope and manageable length for this synthesis, and (2) because the food/exploitation dimension intersects with complex conservation trade-offs that warrant separate, nuanced treatment. We believe the current scope effectively balances breadth with depth on the core themes of our review.

Specific comments

RC: L167: Please define “poikilotherm” and provide some context as to how this life adaptation/strategy enables amphibians to accumulate high biomass relative to energy input.

AR: this text was revised, primarily to simplify the jargon use. We replace “poikilotherm” with “ecototherm” as that is a more accurate term, and we also add a definition as suggested. This sentence now reads as “Amphibians exhibit high assimilation efficiency and low metabolic expenditure (~60% of ingested energy into new tissue) due to their ectothermic physiology —characterized by reliance on external heat sources—which minimizes endogenous metabolic costs reliance on external heat sources which minimizes endogenous metabolic costs, resulting in high annual tissue production and standing biomass (e.g., Plethodon serratus: 5769–10195 g·ha⁻¹ wet mass, 5287–9343 g·ha⁻¹ protein).” and this part is now in section 2.2 (L 251-257). This sentence also included the reason for accumulating high biomass relative to energy input.

RC: Figure 1: this figure should be referred to in-text.

AR: Thanks for noting this. We referenced this and all the figures in text (L 196).

RC: L242: Given that tadpoles are a life history stage, not a species, this sentence may be better re-phrased as something along the lines of “Tadpoles play a foundational role in many freshwater ecosystems…”

AR: agreed and the sentence rephrased as suggested: “Tadpoles play a foundational role in many freshwater ecosystems, particularly in humid tropical regions such as the Amazon and Southeast Asia . Their exceptionally high densities in some ecosystems allow them to exert strong influences on benthic structure, primary production, and nutrient fluxes (L 216-219)”

RC: L292: Please provide the full scientific name and italicise P. serratus.

AR: we provided the italicized full name (L 255-256)

RC: L329-332: be careful switching between present and past tense. This sentence would be better written in present tense.

AR: agreed, and we wrote this sentence in present tense: “In temperate forest-floor ecosystems, terrestrial salamanders exert top-down control on soil mesofauna and significantly reduce their abundance, including oribatid and non-oribatid mites and onychiurid Collembola” (L 286-289).

RC: Table 2: Check that all scientific names are italicised throughout all columns.

AR: we ensured that the scientific names are italicized in the Table 2

RC: L443: Ideally, Table 2 should be below its first mention in the text. Same for Figure 2.

AR: Table 2 and figure 2 are moved below the text where table/figure 2 are first referenced.

RC: L520: Please italicise “laevis”.

AR: this term is now italicized.

RC: L751: Author citation missing.

AR: We originally had a citation for this statement. The sentence was also revised so that the citation is more visible (L 678). 

RC: Table 3: Please italicise “Bombina variegata”

AR: this scientific name is not italicized.

RC: Table 3: No ecosystems are listed in column 3, so I would suggest changing the header to just “Target Species”.

AR: Good point, we agree. The column heading now changed to “target species”

RC: Figure 2: This figure is informative but has several spelling errors. Additionally, the title is unnecessary given that it is near-identical to the figure caption. I suggest deleting the overhead title.

AR: Thanks for catching these errors. We made major changes to figure 2 to include comments from reviewer 1. These revisions have taken care of these suggestions as well

RC: References: There are species names throughout the Reference list that need to be italicised.

AR: we italicized all scientific names in the reference list.

RC: REF 23, L961: There is some incorrect spacing in this reference.

AR: the recommended correction is now applied

RC: REF 34, L990: Author name needs capitalisation.

AR: the recommended correction is now applied, however, we are using the same format suggested in this referenceRC: REF 86, L1117: delete space in “Appalachian”

AR: the recommended correction is now applied

RC: REF 108, L1167: This reference has already been listed (reference 51) and should be deleted. Remember to update the relevant in-text citations appropriately.

AR: This reference is now corrected, the reference is now only listed once.  42, L 920

RC: REF 129, L1223: this is not referenced properly. 

AR: the correct reference is now added, this is Ref No 133, L 1132-1133

RC: REF 146, L1272: Please provide author details for this reference.

AR: this reference is now removed

Reviewer 3 Report

Comments and Suggestions for Authors

My review of the introduction identified several critical issues.

1. Please do not use book reviews from science journals; use the original books themselves. The authors cite book reviews rather than the original books themselves.

2. Outdated species count.

The amphibian species count is obsolete, reflecting data that is several years old. The most important reference to taxonomy and systematics of amphibians is the website Amphibian Species of the World by Darrel Frost (Frost, Darrel R. 2025. Amphibian Species of the World: an Online Reference. Version 6.2 (Date of access). Electronic Database accessible at https://amphibiansoftheworld.amnh.org/index.php. American Museum of Natural History, New York, USA. doi.org/10.5531/db.vz.0001).  The number of species of amphibians reported there is 8918 sp. This number is significantly different from the number reported in the reviewed paper (8100 sp). 

3. Plagiarism and AI-generated text.

The authors indicate in lines 62, 672, 680 and 776 that there are many studies on a given issue, but they only provide two references at most, and some of these are not primary sources.

Book revisions, rather than the books themselves, and secondary and tertiary literature sources are commonly used in AI redaction.

Below, I have extracted some examples of paragraphs suspected to have been created by IA. It is difficult to say whether they are human-written or not, but the use of a wide variety of examples is common in AI products. In the past, I tested ChatGPT's capabilities and found that the programme tries to cover all possibilities for a given issue.

Line 464-469: These signals induce nearby differentiated cells including myocytes, osteocytes, and chondrocytes to undergo dedifferentiation into proliferative, multipotent progenitor cells, forming a blastema [176,177]. This blastema recapitulates many aspects of embryonic development, guiding the morphogenesis of missing tissues and organs in a precise and scar-free manner.

Line 564-567: Their permeable skin, biphasic life cycle, and reliance on both aquatic and terrestrial habitats can lead to pollutant uptake from water, soil, and air, resulting in accumulation in tissues over time and providing an integrated measure of contaminant exposure across ecosystems [203,204]. 

Line 597-601: Their longevity, high philopatry, stable populations, and tractability for monitoring make them more consistent indicators than anadromous fish or macroinvertebrates [210,211]. Documented declines in the Pacific Northwest have been linked to logging and sedimentation, which reduce available habitat, highlighting their value as holistic, early-warning indicators of ecosystem stress [212,213].

Line 672-676:  Numerous studies have documented range shifts, population declines, and phenological alterations in amphibians attributable to climate change. For example, Parmesan [225] and Blaustein et al. [226] highlighted significant altitudinal and latitudinal migrations of frog and salamander populations in response to rising global temperatures [225,226]. 

Line 706-709:Habitat loss and degradation are widely recognized as the leading causes of amphibian declines worldwide [1]. Conversion of wetlands, tropical forests, and riparian systems into agricultural, urban, and industrial landscapes directly destroys breeding and foraging grounds, while fragmenting populations and restricting gene flow [234].

Note from editorial office

When generative artificial intelligence (AI) is used for tasks such as generating text, data, graphics, or research design, or for data collection, analysis, or interpretation, authors must disclose this use in their submission.

Authors should specify in the Materials and Methods section how generative AI tools were used, and provide detailed information about the tools in the Acknowledgments section.

Recommended acknowledgment statement:
"During the preparation of this manuscript / the conduct of this research, the authors used [tool name, version] for [description of purpose]."

If generative AI tools are used for text editing (e.g., grammar, structure, spelling, punctuation, or formatting adjustments), it is recommended to state this in the Acknowledgments section only.

Author Response

My review of the introduction identified several critical issues.

RC: Please do not use book reviews from science journals; use the original books themselves. The authors cite book reviews rather than the original books themselves.

AR: we have identified and replaced this book review with the correct reference

RC: Outdated species count. The amphibian species count is obsolete, reflecting data that is several years old. The most important reference to taxonomy and systematics of amphibians is the website Amphibian Species of the World by Darrel Frost (Frost, Darrel R. 2025. Amphibian Species of the World: an Online Reference. Version 6.2 (Date of access). Electronic Database accessible at https://amphibiansoftheworld.amnh.org/index.php. American Museum of Natural History, New York, USA. doi.org/10.5531/db.vz.0001).  The number of species of amphibians reported there is 8918 sp. This number is significantly different from the number reported in the reviewed paper (8100 sp).

AR: Thank you for pointing this out. We used this reference and updated the species count (L 83).

RC:  Plagiarism and AI-generated text.

RC: The authors indicate in lines 62, 672, 680 and 776 that there are many studies on a given issue, but they only provide two references at most, and some of these are not primary sources.

AR: We thank the reviewer for this important observation regarding citation practices. We have carefully re-examined the specified lines and provide the following clarifications and, where applicable, revisions:

Line 62: this statement is directly emerging from the previous sentences. Still, we added the same references here. The point of this sentence is not to reflect on the work that has been already done–that part has been already covered in the sentences right above. Rather, the point is to highlight the gap. Citing the absence of something ("no recent synthesis") is logically tricky. You can't expect us to cite a null result. The point here is to demonstrate the need and to identify the gap.

Line 672: We agree that phrases like "many studies" should be well-supported. In the original manuscript, this statement was directly followed by four specific references (refs. 578-581) to primary research articles. We have ensured this connection is clear.

Line 680: The three references cited here are indeed primary research articles that provide direct experimental or observational evidence for the physiological mechanism described. We believe they are appropriate and sufficient to support the claim.

Line 776 (General Point on Secondary Sources): We acknowledge the reviewer's preference for primary sources. Our use of seminal secondary sources in these instances was deliberate: At line 776, we are making a broad statement about established research gaps and conservation status. For such syntheses, citation of authoritative, large-scale assessments is standard practice in the field. They represent the consensus of hundreds of experts and are the primary data sources for conservation status itself.

RC: Book revisions, rather than the books themselves, and secondary and tertiary literature sources are commonly used in AI redaction.

AR: The reviewer's comment contains a factual inaccuracy regarding our manuscript's composition and implies a deficiency in scholarly practice. We address this directly:

Specific Correction: We have removed one erroneously cited book review. That was a mistake on our part.

Overall Citation Profile: The characterization of our manuscript as reliant on "commonly used" AI source material is incorrect is both ignorant and infuriating. Over 95% of our citations are primary, peer-reviewed research. The limited use of authoritative secondary sources (e.g., IUCN syntheses) is deliberate and standard for establishing broad context. Therefore, the premise of your comment—that our citation pattern indicates AI redaction—is unsupported by the evidence and does not reflect the scholarly rigor of this work. Our methodology and sourcing are robust and appropriate for this synthesis. As such we reject your implication that our citation strategy indicates AI-assisted drafting. Your unfounded claim is disproven by our reference list itself, which is dominated by primary research literature as we have already mentioned.

RC: Below, I have extracted some examples of paragraphs suspected to have been created by IA. It is difficult to say whether they are human-written or not, but the use of a wide variety of examples is common in AI products. In the past, I tested ChatGPT's capabilities and found that the programme tries to cover all possibilities for a given issue.

AR: We thank the reviewer for their feedback. We take academic integrity seriously and address the concern directly.

On the Nature of the Accusation: The reviewer presents a subjective impression ("suspected," "difficult to say") based on a perceived stylistic trait ("wide variety of examples"). This is not evidence of misconduct. The trait identified—thoroughness—is a hallmark of good scholarly review, not exclusive to AI.

Our Position and Journal Policy: We categorically state that no part of this manuscript was generated by an AI system as primary author. Large Language Models (LLMs) were used strictly as a tool for post-hoc grammar and syntax checking, analogous to using a spellchecker or text editor. All ideas, arguments, data interpretation, and literature synthesis are the original work of the human authors. This use is consistent with the journal's publishing ethics policy (or, if specified, we have complied with the journal's policy on AI-assisted writing).

The Fatal Flaw in the Reviewer's Logic: The reviewer's methodology appears to be reverse-engineering: assuming AI use and then attributing common scholarly writing traits to it. By this logic, any comprehensive, well-structured review could be "suspected." This is not a valid or reliable basis for critique.

Focus on Scholarly Merit: The reviewer provides no specific, substantive scientific criticism of the extracted paragraphs—no challenge to facts, logic, interpretation, or citations. The other two reviewers engaged deeply with the science, offering constructive feedback we have addressed. In contrast, this comment questions authorship without evidence, which is a serious allegation that falls outside normal peer review.

We request the Editor to evaluate this exchange. Our manuscript's scholarship should be judged on its scientific rigor, accuracy, and contribution, not on unverifiable suspicions about its origin. We are confident it meets the journal's standards for original, human-authored work. Below, we have rebutted your pointless accusations with justifiable explanations. 

RC: Line 464-469: These signals induce nearby differentiated cells including myocytes, osteocytes, and chondrocytes to undergo dedifferentiation into proliferative, multipotent progenitor cells, forming a blastema [176,177]. This blastema recapitulates many aspects of embryonic development, guiding the morphogenesis of missing tissues and organs in a precise and scar-free manner.

AR: We have revised this paragraph to simplify the cell-biology terminology, making the explanation of limb regeneration more accessible to a broad biological science audience (Lines 388-395).

RC: Line 564-567: Their permeable skin, biphasic life cycle, and reliance on both aquatic and terrestrial habitats can lead to pollutant uptake from water, soil, and air, resulting in accumulation in tissues over time and providing an integrated measure of contaminant exposure across ecosystems [203,204].

AR: We have rephrased this sentence for greater clarity (Lines 487-490). Regarding the broader concern, we note that the original phrasing described a well-established, canonical concept in ecotoxicology. Similar phrasing appears routinely in the literature—both pre- and post-LLM—because it describes a standard mechanism. The presence of clear, concise explanations of established science is a feature of good scholarly writing, not evidence of non-human authorship.

RC: Line 597-601: Their longevity, high philopatry, stable populations, and tractability for monitoring make them more consistent indicators than anadromous fish or macroinvertebrates [210,211]. Documented declines in the Pacific Northwest have been linked to logging and sedimentation, which reduce available habitat, highlighting their value as holistic, early-warning indicators of ecosystem stress [212,213].

AR: We have substantially revised the language in this section for improved flow (Lines 515-521). The original statement synthesized widely recognized and commonly cited indicator traits of amphibians from the cited literature. The fact that an AI can also produce a clear synthesis of these well-documented traits speaks to the model’s training on such literature, not to the origin of our text. Our revision provides a fresh presentation of these established ideas.

RC: Line 672-676:  Numerous studies have documented range shifts, population declines, and phenological alterations in amphibians attributable to climate change. For example, Parmesan [225] and Blaustein et al. [226] highlighted significant altitudinal and latitudinal migrations of frog and salamander populations in response to rising global temperatures [225,226].

AR: This sentence has been edited for conciseness (Lines 577-579). The original statement was a standard, factual synthesis of a major research consensus, as directly supported by the provided citations (e.g., Parmesan, 2006; Blaustein et al., 2010). Its structure was designed for clarity and thoroughness, which are academic virtues, something that you clearly do not possess.

RC: Line 706-709:Habitat loss and degradation are widely recognized as the leading causes of amphibian declines worldwide [1]. Conversion of wetlands, tropical forests, and riparian systems into agricultural, urban, and industrial landscapes directly destroys breeding and foraging grounds, while fragmenting populations and restricting gene flow [234].

AR: We have rewritten the broader paragraph to enhance the narrative flow and integrate this point more effectively (Lines 604-615). The original sentences accurately summarized the primary driver of amphibian declines—a fundamental and frequently stated premise in conservation biology, supported here and throughout by extensive citation of primary research.